



# A machine learning approach to quantify meteorological drivers of recent ozone pollution in China

Xiang Weng[1], Grant L. Forster[1,2], Peer Nowack[1,3]

[1] School of Environmental Sciences, University of East Anglia, Norwich, NR47TJ, UK
[2] National Centre for Atmospheric Sciences, University of East Anglia, Norwich, NR47TJ, UK
[3] Grantham Institute – Climate Change and the Environment, Department of Physics, and the Data Science Institute, Imperial College London, London SW7 2AZ, UK

*Correspondence to*: Xiang Weng (x.weng@uea.ac.uk)

**Abstract.** Surface ozone concentrations have been increasing in many regions of China for the past few years, in contrast to policy-driven declines in other key air pollutants such as particulate matter. While the central role of meteorology in modulating ozone pollution is widely acknowledged, its quantitative contribution remains highly uncertain. Here, we use a data-driven machine learning approach to assess the impacts of meteorology on surface ozone variations in China for the years 2015 to 2019, considering the months of highest ozone pollution from April to October. To quantify the importance of various meteorological driver variables, we apply non-linear random forest regression (RFR) and linear ridge regression (RR) to learn relationships between meteorological variability and surface ozone in China, and contrast the results to those obtained with the widely used multiple linear regression (MLR) and stepwise MLR. We show that RFR outperforms the three linear methods when predicting ozone using only local meteorological predictor variables. This implies the importance of non-linear relationships between local meteorological factors and ozone, which are not captured by linear regression algorithms. In addition, we find that including non-local meteorological predictors can further improve the modelling skill of RR, particularly for Southern China, highlighting the importance of large-scale meteorological phenomena for ozone pollution in that region. Overall, RFR and RR are in close agreement concerning the leading meteorological drivers behind regional ozone pollution. For example, we find that temperature variations are the dominant meteorological driver for ozone pollution in Northern China (e.g., Beijing Tianjin Hebei region), whereas variations in relative humidity are the most important factor in Southern China (e.g., Pearl River Delta). Variability in surface solar radiation modulates photochemistry but was not considered as such in previous controlling factor analyses, and is found to be the most important predictor in the Yangtze River Delta and Sichuan Basin regions. In general, our analysis underlines that hot and dry weather conditions with high sunlight intensity are strongly related to high ozone pollution across China. This further validates our novel approach to quantify the central role of meteorology. By contrasting our meteorological ozone predictions with ozone measurements between 2015 and 2019, we estimate that almost half of the observed ozone trends across China might have been caused by meteorological variabilities on average. We highlight that these insights are of particular importance given possible increases in the frequency and intensity of weather extremes such as heatwaves under climate change.



## 1 Introduction

Surface ozone is an air pollutant that can induce severe harm to both human health and ecosystems (Lefohn et al., 2018; Lelieveld et al. 2015). In the troposphere, it is primarily produced through photochemically induced reaction chains involving volatile organic compounds (VOCs), nitrogen oxides ($NO_x$) and carbon monoxide (CO) (Monks et al., 2015; Jacob, 2000). Over the last decade, Chinese policymakers have been successfully implementing air pollution control policies and strategies, such as The Clean Air Action Plan in 2013 (Chinese State Council, 2013), to reduce harmful air pollutants. As a result, annual mean concentrations of fine particulate matter ($PM_{2.5}$) have been reduced by 30% to 50% from 2013 to 2018 in China, alongside significant decreases in emissions of ozone precursors such as $NO_x$ and CO (Zhai et al., 2019; Zheng et al., 2018). Despite decreasing trends in $NO_x$ and CO, summertime surface ozone concentrations have been increasing from 2013 to 2019 at a rate of about 1.9 ppb $yr^{-1}$ on average across China, with a faster rate of 3.3 ppb $yr^{-1}$ in the North China Plain (Li et al., 2020).

It is well-known that the effectiveness of ozone production is strongly dependent on the atmospheric chemical regime (e.g., Squire et al., 2015, Archibald et al., 2020), in which ozone production is mainly controlled by the abundance of $NO_x$ or VOCs. Many urban and industrial regions in China have been identified and categorized as being within the VOC-limited regime (Ou et al., 2016; Wang et al., 2017). Under these circumstances, surface ozone reductions may require tighter controls on VOCs emissions together with continuous reductions in $NO_x$, while significant reductions in $NO_x$ emissions without simultaneous and adequate controls on VOCs could lead to increased ozone pollution in the short term (Wang et al., 2019), which could largely explain the recent increases in surface ozone across China. Another factor might have been the large reduction in $PM_{2.5}$, especially during the period of 2013 to 2017, because fewer particles could reduce the aerosol sink of ozone-producing radicals such as hydroperoxyl ($HO_2$) (Li et al., 2019a). However, it is likely that this effect has become less important as $PM_{2.5}$ concentrations continue to decline (X. Chen et al., 2021; Li et al., 2019b).

In conjunction with the effects of changing ozone precursor emissions, the effect of meteorological conditions on ozone concentrations should always be considered. Previous work has identified that ozone variations are strongly co-determined by meteorological factors such as incoming solar radiation, temperature, humidity, atmospheric stagnation, and precipitation (e.g., Otero et al., 2018; Zhang et al., 2018; Lu et al., 2019a). For example, solar radiation is pivotal to the photochemical production and destruction of ozone (Finlayson-Pitts and Pitts, 2000). Higher surface temperatures, and in general tropospheric temperatures, change the chemical reaction rate of many ozone-relevant chemical reactions and will affect biogenic emissions of VOCs such as isoprene and monoterpenes which are also important ozone precursors (Lu et al., 2019a; Doherty et al., 2013; Guenther et al., 1993; Xie et al., 2008; Archibald et al. 2020).Work by Lu et al. (2019b) further indicated that hotter and drier weather conditions were the main drivers for background ozone increase in 2017 in major city clusters of China. Similarly, Ma et al. (2019) suggested that high biogenic VOCs emissions and meteorological conditions indicative of heatwaves such as high temperature, low wind speed and no precipitation can elevate ozone pollution in the North China Plain (NCP). Furthermore, studies by Wang et al. (2021) and Pu et al. (2017) also found enhanced ozone concentrations during heatwaves



in the Pearl River Delta (PRD) and Yangtze River Delta (YRD). Such links between meteorology and ozone pollution provide
clear evidence for the necessity to quantify the influence of meteorological factors or even climate change on ozone pollution
in China (e.g., Lu et al., 2019a; Meehl et al. 2018). Characterizing the major meteorological drivers behind ozone variations
in different regions of China will also be crucial for achieving effective mitigation of ozone pollution now and under future
changes in climate.

To quantify the importance meteorological drivers, previous studies such as Li et al. (2019a) and Han et al. (2020) adopted

stepwise multiple linear regression (MLR) to derive linear relationships between meteorological factors and measured surface
ozone concentrations across China. Both of these studies demonstrated the significant skill of stepwise MLR in modelling
ozone and in quantifying the driver-response relationships. Nevertheless, a key limitation of stepwise MLR or conventional
MLR is that these methods are not able to accurately capture non-linearity, which is a severe constraint given that non-linear
relationships between meteorological factors and ozone, e.g., between temperature and ozone, are to be expected (e.g., Pu et
al., 2017; Gu et al., 2020; Archibald et al., 2020). In addition, MLR can suffer from severe loss in predictive skill and reliability
in settings where a large number of (collinear) meteorological factors are considered as predictors (cf., the curse of
dimensionality in high-dimensional regression problems; Nowack et al., 2021; Bishop, 2006). Although the stepwise MLR
approach adopted by Li et al. (2019a) can overcome collinearity and overfitting to some extent because only a few predictors
that are particularly strongly influencing ozone concentrations are kept, it is inevitable that many relevant meteorological
factors will be excluded from the final MLR predictions using that approach.

In order to capture non-linear relationships between many meteorological factors and ozone and to overcome the potential

limitations of considering collinearity and high-dimensional settings in MLR, we will use a machine learning approach as the
next logical step to advance such controlling factor analyses of ozone pollution. Specifically, we will adopt random forest
regression (RFR) (e.g., Grange et al., 2018; Stirnberg et al., 2021) as a non-linear approach and contrast the results to a linear
statistical learning approach that is also robust in high-dimensional settings in the form of ridge regression (RR) (e.g., Nowack
et al., 2018). Both RFR and RR will also be compared with more conventional statistical methods such as MLR and stepwise
MLR.

Our paper is structured as follows. In Sect. 2, we describe the data used in this study and the modelling framework of the

two machine learning algorithms, namely, RFR and RR. In Sect. 3, the performances of RFR and RR will be discussed first
and then compared to those achieved with MLR and stepwise MLR. Afterwards, we summarize the most important
meteorological drivers for surface ozone as identified by RFR and RR. Finally, we conduct a trend analysis of recent surface
ozone changes in China, and use our method to estimate the contribution of meteorological effects.



## 2 Methods

### 2.1 Surface ozone and meteorological data

The surface air quality measurement data used in this study were obtained from https://quotsoft.net/air/ (Wang, X. L., 2021; last accessed: 13 July 2021) which is a mirror of the data from the China Ministry of Ecology and Environment (MEE). For the purposes of quantifying ozone pollution severity, we use the maximum daily 8-hour rolling mean (MDA8) ozone calculated following the guidelines by the Ministry of Environmental Protection of People's Republic of China (MEP, 2012). The calculation selects the maximum value from 8-hour rolling means of ozone for each station between 08:00 and 24:00 on each day. To be considered, each station must have a valid 14 hours data record of 8-hour rolling means ozone within 08:00 to 24:00 on the respective day, otherwise MDA8 ozone is not calculated for that day. Previous studies (e.g., Li et al., 2020; Li et al., 2019a; Han et al., 2020) have focused on ozone pollution during the boreal summer months i.e., June, July, and August (JJA) as the season with the most frequent occurrence of extreme ozone episodes in China. In this work, we extend this analysis period to include the months from April to October to account for the fact that the seasonality of ozone does not follow a uniform pattern across China. For example, peak ozone concentrations are often found during autumn over the PRD region (Gao et al., 2020; see Fig. S1 in the Supplementary Material). In addition, we further constrain our analysis to the period 2015 to 2019 to maintain greater consistency of the ozone data throughout our analysis period as the MEE included far fewer measurement stations prior to 2015. In order to maintain consistency and reliability of all ozone data from stations within the study period, only those stations with over 80% temporal coverage of MDA8 ozone data record in each year are selected. For quality assurance of the data, we further examined each station's MDA8 ozone variation individually and noticed that measurements from some stations appeared to show a less reliable data record than others. This was for example evident from extended periods of non-fluctuating ozone levels (see Fig. S2), or from sudden unusual MDA8 spikes, usually followed by periods of suppressed ozone variability (see Fig. S3). According to our best judgement, such abrupt changes or unrealistically low variability are unlikely to reflect actual ozone pollution profiles. Data from stations that showed such unusual time evolutions were excluded from our analysis as to avoid the inclusion of unrealistic artefacts. The list of stations that are not used in this study is summarized in Table S1.

To study regional meteorological drivers of ozone, we distinguish four regions of particularly high population density known as Beijing-Tianjin-Hebei (BTH, which is equivalent to north China plain), Yangtze River Delta (YRD), Pearl River Delta (PRD) and Sichuan Basin (Sichuan), using definitions frequently used in previous studies (e.g., Li et al., 2019a; Han et al., 2020). The boundaries of these four regions are adjusted to ensure that stations in each region have similar topography and equivalent elevation. The four regions are also known as the target areas for air pollution reduction in Chinese government plans (MEE; http://www.mee.gov.cn/hjzl/dqhj/cskqzlzkyb/ last access: 1 December 2021; Li et al., 2019a). The locations of stations within the four regions are indicated by red dots in Fig. 1.

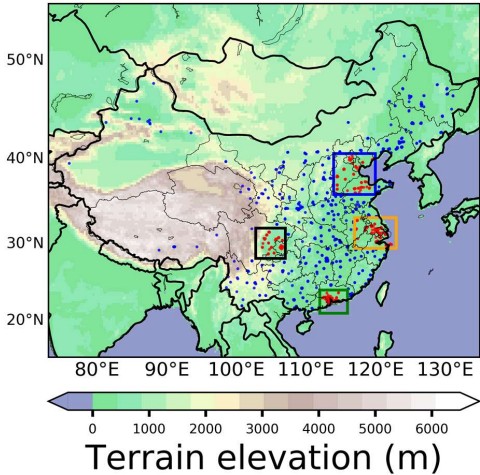

**Figure 1. Elevation height (m) and locations of all ground-based stations and the four megacity cluster regions, BTH (blue box; 114° E-120° E, 36° N-40.62° N), YRD (orange box; 117° E-123° E, 29.458° N-33.238° N), PRD (green box;112° E-116° E, 21° N-24.111° N), Sichuan Basin (black box; 102.8° E-107.061° E, 28.2° N-31.976° N). Red (blue) dots indicate locations of stations within (outside) the four regions.**

For the meteorological data, we use the gridded ERA5 reanalysis product (Hersbach et al., 2020) available at https://cds.climate.copernicus.eu/ (last accessed: 11 November 2021). Specifically, we use hourly data for a total of 11 meteorological variables at 0.25°×0.25° spatial resolution, namely, temperature at 2 m (T2), boundary layer height (BLH), mean sea level pressure (SLP), surface solar radiation downward (SSRD), relative humidity at 1000 hPa (RH), total precipitation (TP), zonal wind at 10 m (U10), meridional wind at 10 m (V10), zonal wind at 850 hPa (U850hPa), meridional wind at 850 hPa (V850hPa) and vertical velocity at 850 hPa (W850) for the same time period as for the ozone station data. Then the MDA8 ozone data are spatially averaged within the dimensions of each ERA5 grid cell to obtain the best possible spatial match between the station-based ozone data and the large-scale meteorological factor data.

The variables of T2, BLH, SLP, RH, TP, U10, V10 can also be found as predictors in the multi linear regression (MLR) studies of Han et al. (2020) and Li et al. (2019a). Surface solar radiation downward (SSRD) is included in this study instead of adding a cloud coverage term as done by Han et al. (2020) and Li et al. (2019a). Essentially, we consider SSRD to more directly characterize the local photochemical environment for ozone production and loss than cloud coverage. Zonal and meridional wind at 10 m may be important for dispersion of ozone's precursors on a local scale. Both zonal and meridional wind at 850 hPa are adopted in this study in order to encompass the effect of transport of more polluted or cleaner air from remote regions. Wind at 850 hPa is less likely to be affected by orography than wind at 10 m altitude, and it is thus better suited for considering the effect of larger scale transport and dispersion. Additionally, we represent the role of vertical transport of air masses by including vertical velocity at 850 hPa as another factor.





**2.2 Data pre-processing**

Prior to modelling ozone, we pre-processed the meteorological data by averaging the raw hourly data over different periods each day and this process is summarised in Table 1. The averaging periods were not the same for all meteorological variables. For example, T2, SSRD, SLP, RH, and W850 are averaged between local time (UTC+8:00) 06:00 to 18:00 on each day. The average of these hours is sufficient to cover all daytime hours when ozone is photochemically produced from April to October. Total precipitation is calculated as the sum of accumulated precipitation for all hours from 06:00 to 18:00. For zonal and meridional wind at 10 m and 850 hPa, data are averaged over 06:00 to 12:00, which covers the main hours that may have potential fresh emission of precursors and transport or dispersion of precursors or ozone. Boundary layer height (BLH) is averaged over 00:00 to 12:00 for the consideration of both potential night-time emission of industrial activities when boundary layer is still low and transportation emission during morning rush hours. Through this process, raw hourly meteorological data can be converted to daily format, temporally matching with MDA8 ozone data.

**Table 1. Summary of the meteorological controlling factor variables and the respective times of day considered in their averages. The motivation behind each selected time period is provided in the main text. Note: a positive zonal wind means westerly; positive meridional wind means southerly; positive vertical velocity means downward motion.**

| Acronyms | Names and units of variables | Average period |
|----------|------------------------------|----------------|
| T2 | temperature at 2 m (K) | 06:00 to 18:00 |
| SSRD | surface solar radiation downward (J m$^{-2}$) | 06:00 to 18:00 |
| SLP | mean sea level pressure (Pa) | 06:00 to 18:00 |
| RH | relative humidity (%) | 06:00 to 18:00 |
| BLH | boundary layer height (m) | 00:00 to 12:00 |
| U10 | zonal wind at 10m (m s$^{-1}$) | 06:00 to 12:00 |
| V10 | meridional wind at 10m (m s$^{-1}$) | 06:00 to 12:00 |
| TP | total precipitation (m) | 06:00 to 18:00 (sum) |
| U850hPa | zonal wind at 850hPa (m s$^{-1}$) | 06:00 to 12:00 |
| V850hPa | meridional wind at 850hPa (m s$^{-1}$) | 06:00 to 12:00 |
| W850 | vertical velocity at 850hPa (Pa s$^{-1}$) | 06:00 to 18:00 |

Finally, both ozone data and meteorological data are deseasonalized. Specifically, for MDA8 ozone and the converted daily meteorological variables, we first calculate 15-day moving window averages centered on the particular calendar date from 2015 to 2019. We then take the difference between each day's MDA8 ozone or daily meteorological variables and these 15-day averages to obtain daily anomalies, creating smooth time series of deseasonalized MDA8 ozone and deseasonalized meteorological variables.





**2.3 Machine learning methods for modelling MDA8 ozone**
To model the relationships between meteorological variables and MDA8 ozone concentrations in China, we use two
regression algorithms, a non-linear approach known as random forest regression (RFR) and a linear approach called ridge
regression (RR). Within our framework, the predictors are the deseasonalized meteorological variables from ERA5 and the
dependent variable is the deseasonalized ground-based MDA8 ozone. For RR, both the deseasonalized meteorological
variables and the deseasonalized ozone time series are standard-scaled (normalized to zero mean and unit standard deviation)
as to avoid an imbalance of factors in the regularization part of the RR cost function (Nowack et al., 2018).
Both RFR and RR have been extensively described elsewhere (e.g., Nowack et al., 2018; Grange et al., 2018; Mansfield
et al., 2020; Nowack et al., 2021) and it is beyond the scope of this study to provide an in-depth description. Briefly, RFR is
based on learning an ensemble of decision trees, where each individual tree splits data into groups until reaching certain pre-
set definitions for data 'purity' (Breiman, 2001; Grange et al., 2018). RR is a least-squares linear regression method augmented
by L2-regularization with the goal to avoid overfitting in high-dimensional regression settings, especially in regression
problems with strong collinearity (McDonald, 2009). Both RFR and RR are known to handle collinearity comparatively well
(e.g., Dormann et al. 2013), which is key given that many of meteorological variables such as temperature and solar radiation
are correlated with each other. To assess whether these two machine learning algorithms can improve the accuracy of ozone
modelling compared to conventional statistical methods, we will contrast our results to multiple linear regression (MLR) - that
may not be highly capable of handling collinearity and overfitting and stepwise MLR. For MLR, we simply adopt the same
modelling framework of RFR and RR; all 11 meteorological variables are ingested into MLR as predictors. For stepwise MLR,
we adopted a similar approach as Li et al. (2019a): we start with 11 deseasonalized meteorological variables as predictors in
MLR and remove one predictor at a time based on the smallest significance of the regression coefficient in each new subset of
predictors, until there are only 3 meteorological predictors left. These 3 predictors are considered to be important predictors
and are used in the final model of stepwise MLR for modelling deseasonalized MDA8 ozone.
**2.4 Training, testing and cross-validation in machine learning**
Supervised machine learning approaches such as the two algorithms we use here involve distinct training, validation and
testing phases to tune the relevant hyperparameters (explained in detail below) and to validate the skill of the resulting
predictive functions on new, unseen data not used in the training and tuning process (e.g., Bishop, 2006). During the training
phase, both predictors (i.e., deseasonalized meteorological variables) and dependent variable (i.e., deseasonalized MDA8
ozone) are available and each machine learning regression algorithm is fit to this dataset, assuming different combinations of
values for the hyperparamters of each algorithm. The best objective combination of hyperparameters is then found in the
validation step by predicting ozone values for a validation dataset not used at the training stage (e.g., for a different year in the
data record). During the testing phase, the trained and validated algorithm is used operationally to make new predictions for
ozone values given a new dataset for the meteorological variables as input to the machine learning function. These test set



predictions can then be used to measure the "out-of-sample" skill of the algorithm in predicting ozone pollution given certain
meteorological conditions. In this study, we split the 5-years of data (2015 to 2019) systematically into training/validation and
testing datasets one at a time and in a rotating fashion. Specifically, 4 of these 5 years are classified as training/validation data,
leaving 1 year for testing. To ensure that we are measuring the true predictive performance and relationships, our predictive
results and model evaluations are only conducted for the test data, which has not been used at the training and validation stages.
This process rotates until ozone data for each year have been assigned once as test data so that all 5 years of data can be
predicted by RFR and RR.
Machine learning regressions such as RFR and RR optimize their predictive performance by tuning certain sets of
hyperparameters. To determine the most suitable set of hyperparameters, we use a statistical cross-validation method. Initially,
we split the 5 years of data into 1 test year and 4 training/validation years. For cross-validation, the 4-year training/validation
set is further split into four folds (one year per fold). We then run a grid search over pre-defined combinations of
hyperparameters by training on three folds and predicting on the fourth fold in a classic 4-fold cross-validation procedure. We
finally select the best estimated set of hyperparameters on the basis of the average validation data prediction performance as
measured by the coefficient of determination ($R^2$-score), and refit model coefficients using this set of hyperparameters for the
entire 4 years of training/validation data. We note that we avoid a 'leave-one-out' cross validation method as we expect
autocorrelation in our data (i.e., MDA8 ozone may share similarity in adjacent dates), which, intuitively, could lead to an
overestimate of our predictive skill if testing data immediately follows training data points.
The ranges of hyperparameters we search over for both RFR and RR are set as follows. For RFR, the maximum depth for
trees growing is iterated in a step of 1 from 8 to 15. Maximum percentage of features and maximum samples (with bootstrap
method) are set from 20% to 90% and 30% to 80% with 10% incremental step, respectively. Total tree number for the forest
is set at 200 as a compromise between model complexity and runtime. Optimizations further showed that the minimum samples
per leaf is best set to 3 in our RFRs so that we finally kept this value constant in our grid searches. In terms of RR, the
regularization strength is iterated over a range of 1 to 199 with incremental step of 2, which appeared to encapsulate the best
solution in each case. A detailed explanation of these hyperparameters for RFR and RR is for example provided in Nowack et
al. (2021).

### 2.5 Identifying and quantifying importance of meteorological drivers

Both RFR and RR can enable the identification of the most important meteorological drivers for MDA8 ozone and can
help to quantitatively evaluate their relative importance. For RFR, we here measure the importance of each meteorological
predictor through a metric called Gini importance. A greater Gini importance implies a greater influence of a particular
predictor (i.e., the deseasonalized meteorological variable) on the dependent variable (i.e., deseasonalized MDA8 ozone) (e.g.,
Menze et al., 2009; Zhao et al., 2019, Kuhn-Régnier et al., 2021). Since we train the RFR five times for each set of 4-year
training/validation data, we average the Gini importance scores for each meteorological predictor across all five runs for our
final discussion below. For RR, similar to MLR, importance of each predictor is evaluated by the magnitude of each predictor's



averaged slope (linear regression coefficient) across all 4-year training/validation datasets, which represents the linear effect
of each predictor onto ozone, given that all predictors are standard-scaled (see Sect. 2.3).
**3 Results and discussion**
**3.1 Machine learning performances for modelling ozone using local meteorological predictors**
It is important to first assess how well of these machine learning algorithms can model ozone by using only meteorological
variables as predictors. Therefore, we adopt the coefficient of determination ($R^2$) (i.e., the square of Pearson correlation
coefficient, R) as a standard metric for prediction performance on the deseasonalized MDA8 ozone data. As mentioned above,
to measure the true predictive skill of the machine learning functions, we only compare our predictions for out-of-sample test
data that are not used during training/validation stages against the deseasonalized measured MDA8 ozone data.
The predictors used by RFR and RR here are only the local meteorological variables, i.e., each ERA5 grid point's
meteorological variables are used as predictors to model averaged deseasonalized MDA8 ozone for that particular grid location.
The average prediction performance of RFR and RR by comparing predictions across all test years against the deseasonalized
measured MDA8 ozone data across China is illustrated in Fig. 2.

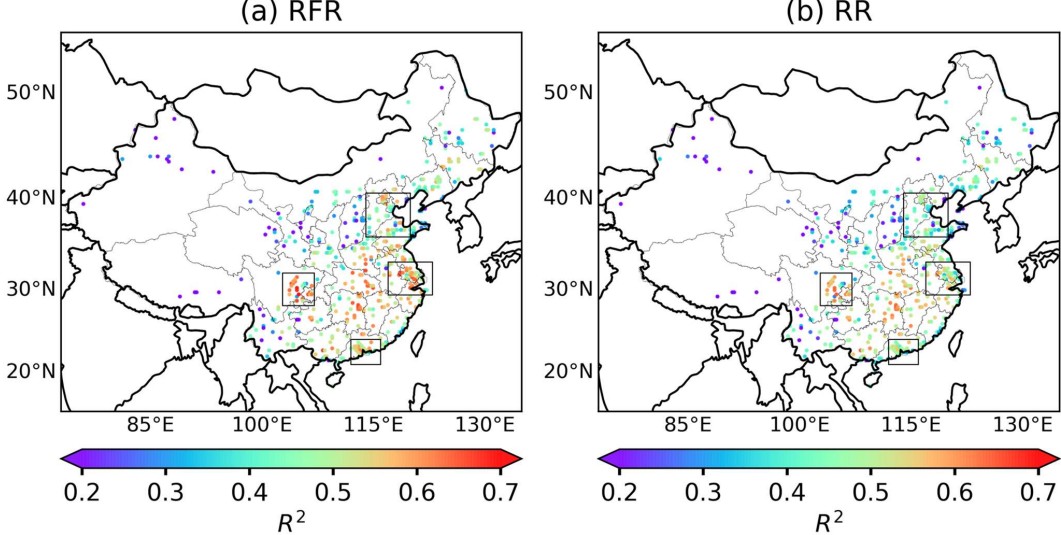


**Figure 2. Coefficient of determination ($R^2$) between deseasonalized observational MDA8 ozone and deseasonalized predicted values**
**in random forest regression (a) and ridge regression (b). The skill is only measured for the respective test datasets. Each dot**
**represents the center of the ERA5 grid location, within which station values for MDA8 ozone are averaged.**



Overall, the model performance of RFR generally surpasses the one of RR over most regions of China, with higher $R^2$
values in grid locations within the Sichuan Basin, YRD, PRD and other regions of southeast China. $R^2$ scores for RFR generally
range from 0.5 to 0.6 across China while RR reaches $R^2$ scores from 0.4 to 0.5. RFR and RR perform similarly over the central
region of BTH, while in the northern region of BTH (e.g., Beijing), $R^2$ values are still found to be higher in RFR than RR. The
averaged $R^2$ across all ERA5 grid locations within BTH, YRD, PRD, and Sichuan Basin is 0.46, 0.56, 0.53 and 0.57
respectively for RFR, which are all higher than the equivalent $R^2$ for RR (BTH: 0.41, YRD: 0.48, PRD: 0.47, Sichuan Basin:
0.53).

In order to examine whether RR can improve the model performance by being less sensitive to collinearity, we also
applied MLR with all 11 meteorological predictors and the stepwise MLR approach with the 3 most important meteorological
factors in the final MLR for comparison (see Sect. 2.3). Overall, stepwise MLR shows the worst performance with $R^2$ scores
ranging from 0.3 to 0.4 across China, with averaged $R^2$ scores in BTH, YRD, PRD and Sichuan Basin at 0.39, 0.45, 0.43 and
0.52, respectively (see Fig. S4b in Supplement for spatial distribution of $R^2$ scores). This suggests that the stepwise MLR
approach may carry a risk of not including all important meteorological predictors in the regression model. However, RR does
not show noticeable improvements over MLR, as evident from similar regionally averaged $R^2$ scores (see Fig. S4a), suggesting
that the problem of collinearity is still limited given the use of 11 meteorological predictors. The enhanced performance of
RFR compared to RR may therefore be attributed to RFR being able to model non-linear relationships between local
meteorological variables and ozone, indicating that a flexible machine learning approach such as RFR that can capture non-
linearity is more suitable to reflect relationships between local meteorological factors and ozone.

### 3.2 Predictive skill using additional non-local meteorological predictors

Meteorological phenomena usually belong into a larger spatial context. For example, high-pressure systems usually take
in a larger spatial domain, suppressing air flow in certain directions. Consequently, it seems intuitive that a meteorological
controlling factor framework for ozone might benefit from including additional non-local information in the regressions, i.e.,
if we were to consider surrounding meteorological context information that is not just limited to the predicted ozone grid point
in question (Ceppi and Nowack 2021).
We thus ran a second version of our controlling factor analysis in which we did not just include local values of
meteorological drivers, but additionally consider a spatially wider effect of meteorology on a two-dimensional (2D) domain
of meteorological variables. This is possible since both RR and RFR are less prone to collinearity and overfitting in high-
dimensional regression settings than simple non-regularized MLR approaches would be, meaning that the additional
information included in the regressions might well outweigh the cost of adding more predictors.
In detail, for each ozone target grid point, we include a meteorological context by adding each meteorological variable
within a 7.5°×7.5° rectangle domain around the center of this target grid point to the set of model predictors, i.e., all the
meteorological variables from the ERA5 0.25° × 0.25° grids within this 7.5°×7.5° rectangular domain are used as individual
predictors in the regression models. This adds potentially important information about the larger-scale meteorological situation





to our predictions, but also significantly increases the dimensionality (number of predictors) of our regression problem and
increases the number of collinear predictors. Indeed, we find that through the additional L2-regularization in RR with 2D
expansion (denoted as RR-2D), its predictions by far outperform its MLR-2D equivalent which now suffers from severe
overfitting (compare $R^2$ scores in Fig. 3a and 3b). Noteworthy, with the increase of dimensionality in RR-2D, the regularization
strength now is adjusted to larger values starting from $10^3$ to $10^9$ with a factor of 1.42 incremental increase at each step, which
is much higher than the regularization strength set in RR with only local predictors. Suh a large increase of range is due to the
consideration of adding large number of meteorological predictors within the 2D domain, and it ensures that the best solution
with the most suitable regularization strength for each run can be well covered within this range. The overall $R^2$ scores for RR-
2D ranges from 0.5 to 0.6 while $R^2$ in MLR-2D ranges from 0.3 to 0.4; MLR-2D is overall worse than MLR with only local
meteorological predictors in terms of $R^2$. It is well-known that RFR may not be as effective at handling multi-collinearity in
very high dimensional settings as RR (e.g., Dormann et al., 2013) and its training time also increases exponentially with the
number of predictors. We thus only ran RFR with 2D expansion (denoted as RFR-2D) for the southern Chinese PRD region,
where we found a particularly large $R^2$-score improvement after including non-local predictors in RR-2D ($R^2$=0.60) as
compared to local RR ($R^2$=0.47), and even non-linear local RFR ($R^2$=0.53). These results highlight the apparent importance
of large-scale meteorological phenomena in this region. However, we find that RFR-2D improves the average $R^2$ score (0.57)
relative to RR and RFR with only local predictors, but does not perform better than RR-2D.

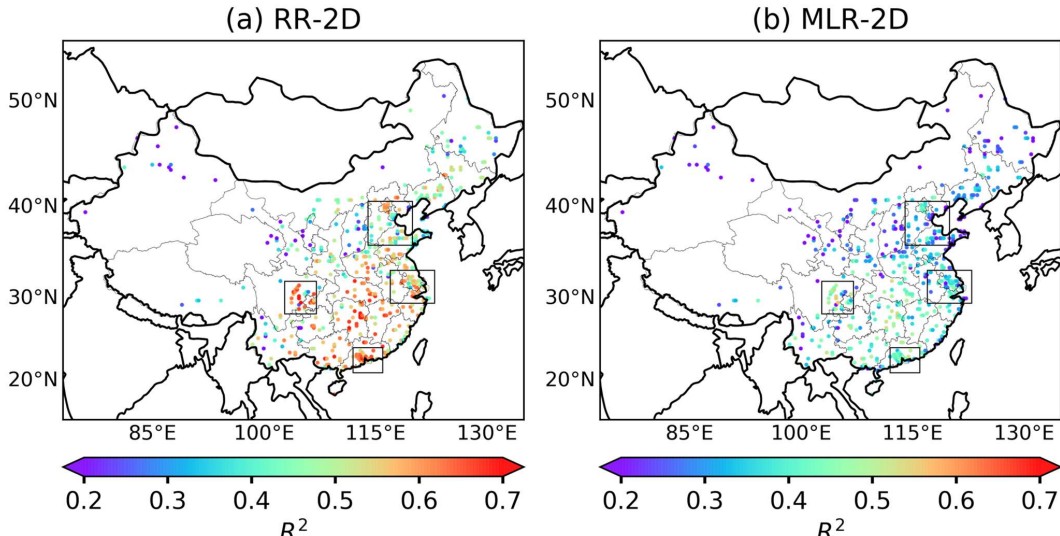


**Figure 3 Coefficient of determination ($R^2$) between deseasonalized observational MDA8 ozone and deseasonalized predicted values**
**of MDA8 ozone in ridge regression (RR) with 2D expansion (a) and MLR with 2D expansion (b).**





For clarity, Table 2 summarizes the averaged $R^2$ in each region by all machine learning methods including RFR, RR,
MLR, stepwise MLR, RR-2D, MLR-2D and RFR-2D. In summary, RFR and RR-2D are overall the two machine learning
algorithms with highest $R^2$ in these four regions, while MLR and RR are equivalent.
**Table 2. Averaged $R^2$ in the four regions by different machine learning algorithms, namely RFR, RR, MLR and stepwise MLR with**
**only local meteorological predictors, RR-2D, MLR-2D with additional two-dimensional (2D) non-local meteorological variables and**
**RFR-2D which is only conducted for PRD region. In general, with only local meteorological variables, RFR performs the best with**
**highest averaged $R^2$ in four regions. RR-2D and RFR-2D show improvement over PRD region compared to RFR.**

| Method | BTH | YRD | PRD | Sichuan |
|---|---|---|---|---|
| RFR | 0.46 | 0.56 | 0.53 | 0.57 |
| RR | 0.41 | 0.48 | 0.47 | 0.53 |
| MLR | 0.41 | 0.48 | 0.47 | 0.53 |
| stepwise MLR | 0.39 | 0.45 | 0.43 | 0.52 |
| RR-2D | 0.47 | 0.54 | 0.60 | 0.58 |
| MLR-2D | 0.31 | 0.35 | 0.42 | 0.43 |
| RFR-2D | - | - | 0.57 | - |

**3.3 Regionally averaged prediction skill**
In order to assess the performance of the algorithms in modelling regional average ozone, we further compared our
regionally-averaged machine learning predictions by RFR, RR and RR-2D against observations for each of the four selected
regions in China (Fig. 4), whereas previously we compared regional averages based on predictions for individual grid points
whose $R^2$ scores were subsequently averaged within each region. For this purpose, we averaged all $0.25° \times 0.25°$ grid point
observations and model results within each region first and then compared the resulting time series for each test dataset directly.
The results of regional averaged predictions and observations for each region are shown in Figure 4, where the goal for the
predictions is to fall as close as possible onto the 1:1-line, in combination with a high $R^2$-score (i.e., square of Pearson
correlation, R). With only local meteorological predictors, RFR still outperforms RR regarding both the coefficient of
determination ($R^2$, same calculation method as above) and slope (closer to 1) in all four regions. This can likely be attributed
to the ability of RFR to capture non-linearity as well.
Using this calculation method, regional $R^2$ are much higher; for RFR, regional $R^2$ in BTH, YRD, PRD and Sichuan Basin
is 0.71, 0.75, 0.7 and 0.83, since each grid is more prone to the effect of local emissions and related local uncertainties as
regional average can factor out the local effects (i.e., emissions and uncertainties) to some extent. For instance, stations that
are located relatively close to emission source may be more influenced by NO titration effect which may lower ozone level
(Sillman, 1999). This effect can be more significant in some urban areas (Li et al., 2017) or stations affected by fresh emission
of NO from power plants (X. Zhang et al., 2021). On the other hand, nearby emission of precursors may also be the dominant
factor in driving ozone in regular weather condition. Given both of these effects, ozone production in these stations may be
less sensitive to meteorological drivers but more influenced by local emissions.

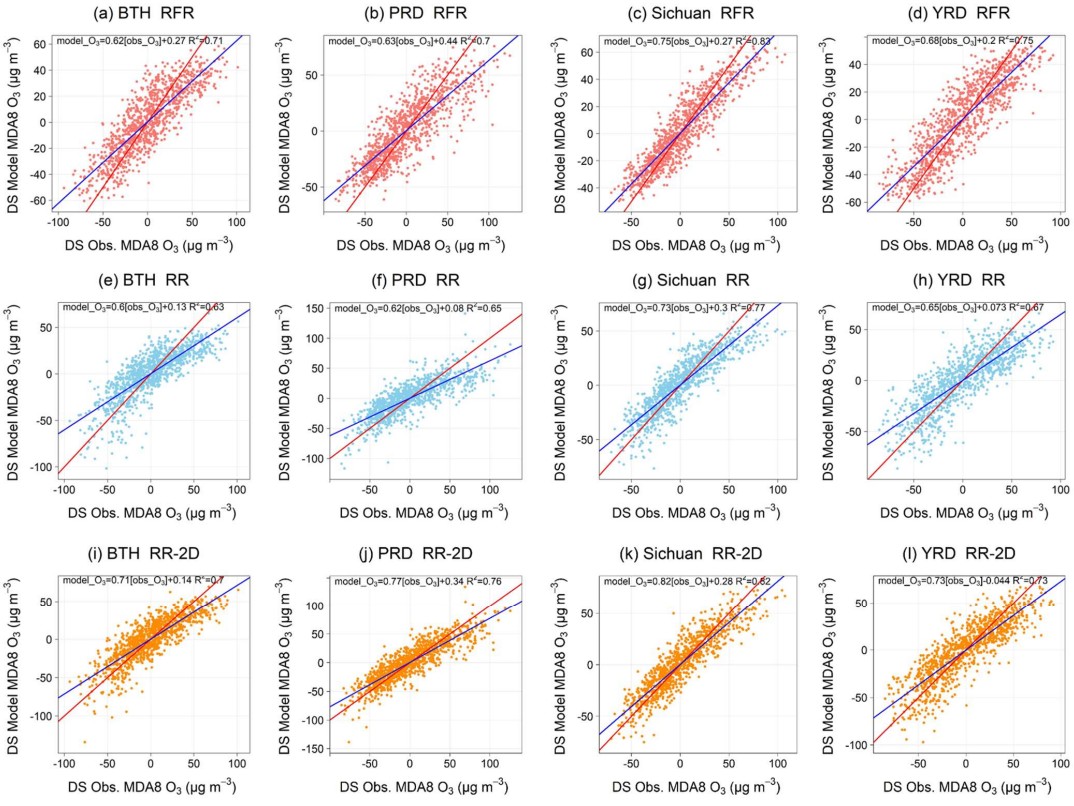

**Figure 4 (a)-(d) Comparison of regional averages of deseasonalized MDA8 ozone between model predictions and observations for RFR, (e)-(h) RR and (i)-(l) RR-2D. Linear fits between predicted and observed data are indicated by blue lines; red lines are the ideal 1:1 lines. The values for both models and observations are averaged over all ERA grid points in each region. Each graph contains information of the linear regression with slope and $R^2$ score (i.e., square of Pearson correlation, R).**

In summary, all three machine learning methods show clear skill in modelling ozone variability driven by meteorological variables. However, similar to results by Han et al. (2020), all linear fits in all regions for both RFR and RR have slopes lower than 1, suggesting a systemic underprediction of ozone for the highest observed ozone levels (higher than the deseasonalized zero mean) and overpredictions of ozone for low ozone pollution regimes (lower than the deseasonalized zero mean). As previously mentioned, such a mismatch may - at least to a degree - arises from non-meteorological factors such as the effect of precursor emissions, which are not taken into account here. Although regionally averaged prediction skill is less affected by local emissions, it will not be completely free from such effects. However, the increase of the magnitude of the slopes in RR-2D with closer to 1 also suggests that considering non-local meteorological variables may help improve the performance of ozone pollution controlling factor analyses, even if non-linearity is not intrinsically taken into account.





**3.4 Quantifying the importance of meteorological predictors**

We next aim to quantify how important each local meteorological predictor is for ozone pollution across China. For RR, we use the regression slope as a standard metric to measure how important of each the meteorological predictor on ozone pollution. A large positive value for the slope (regression coefficient) of a meteorological predictor indicates that the predictor has a strong positive effect on ozone levels and vice versa. Since each of the 4-year training data is learned independently, we will show averaged results. For RFR, we measure each predictor's importance through Gini importance (see Sect. 2.5). The highest absolute value for both the RR slope or RFR Gini importance is interpreted as the most important meteorological driver variable identified through our data-driven learning procedure. Note that Gini importance only allows to measure relative influences of predictor variables on ozone variability, but not the sign of the influence, i.e., a high value of Gini importance score is not able to determine whether the predictor has positive or negative effect on ozone.

The Gini importance scores estimated by RFR and the slopes learned by RR for each region are shown in Fig. 5. Both Gini importance scores and slopes are initially estimated for every ERA5 grid location within each region and then averaged across the entire region and across all five learned regression functions.

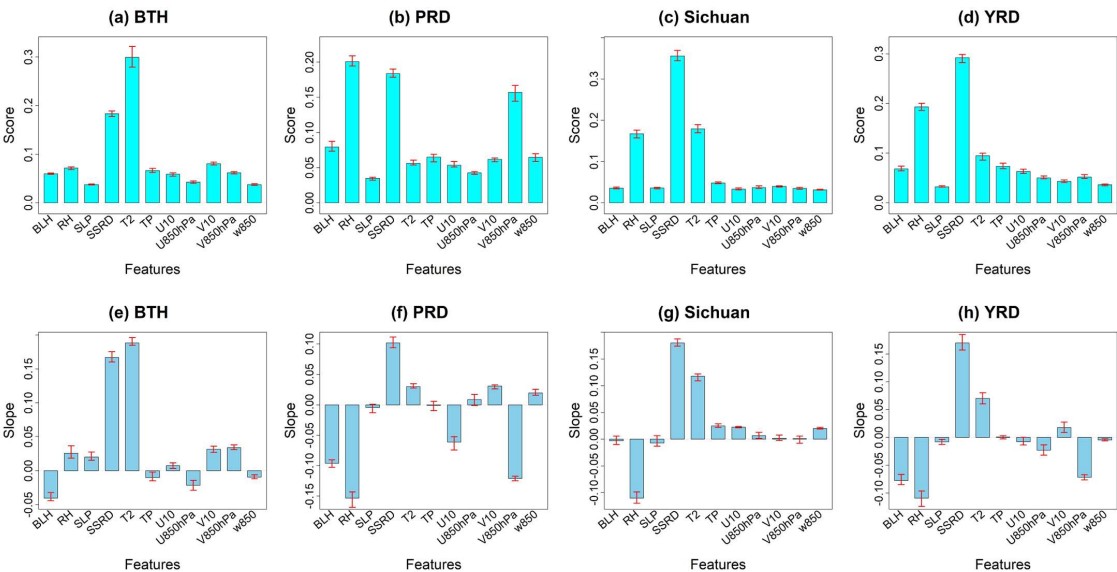

**Figure 5 (a)-(d) Average Gini feature importance scores of each meteorological variable for RFR in each region. (e)-(h) Average slopes of each meteorological variable for ridge regressions in each region. The red bars indicate the range of importance scores/slopes found across the five regression models learned to predict the left-out test years.**

In general, both RFR and RR show good agreement in terms of identifying the dominant meteorological drivers for each region. Temperature at 2 m is found to be the most important meteorological driver for ozone in BTH, followed by surface





solar downward radiation, albeit the relative difference between these two variables differs more clearly for RFR, which might
be caused by non-linearity in the ozone-temperature relationship (Supplementary Fig. S5). Temperature was also identified as
the most important meteorological variable in BTH by Li et al. (2019a) using MLR. Moreover, a more pronounced positive
correlation between daily maximum temperature and MDA8 ozone is found in northern regions of China (Fig. 6a), which is
consistent with the findings of these two machine learning algorithms. Biogenic emissions can be intensified during heatwaves
in BTH, leading increased ozone (Ma et al., 2019). Additionally, high temperature conditions may also lead to the
intensification of certain anthropogenic emissions such as solvent evaporation. A detailed emission inventory in 2013 for BTH
shows that solvent use makes the highest contribution to NMVOC emissions at 46.7% in the industry sector (Qi et al., 2017).
Song et al. (2019) conducted a one-year observation (from April 2016 to March 2017) of VOCs at an urban site in BTH and
found that biogenic emissions and solvent use can make major contribution to ozone formation, and the concentrations of the
reactive VOCs species derived from these sources are found to have a positive correlation with temperature. In summary, with
higher temperature, biogenic emissions and solvent evaporation may be more intense, which may be one of the underlying
causes for elevated ozone pollution in BTH with higher temperatures.

For both YRD and Sichuan, surface solar radiation is most important determinant of ozone variations, with RR slopes
again indicating the expected positive relationship between sunny, clear-sky days and high ozone pollution. Solar radiation is
also found to be important in BTH, PRD by RFR and RR, suggesting its consistent importance across China. The importance
of solar radiation should be given more consideration in assessing the effect of meteorological drivers on ozone pollution.
High solar radiation stimulates the photochemical environment, which has been suggested as one of the key mechanisms in
YRD by Pu et al. (2017). From a large-scale meteorological point of view, such clear-sky conditions in YRD that may enhance
severe ozone pollution in this region are modulated by the western Pacific subtropical high (WPSH) (Shu et al., 2016; Chang
et al., 2019; Shu et al., 2020). In the Sichuan Basin, with complex terrain that can complicate atmospheric circulation, ozone
pollution is often associated with the occurrence of high-pressure systems associated with clear-sky conditions and high
temperatures (Ning et al., 2020), which is also identified by both RFR and RR.

A distinct difference in the weather-ozone coupling relationships is found for PRD, where relative humidity is the
dominant meteorological driver. Specifically, a negative slope of RH in RR suggests that drier conditions are strongly favorable
for peak ozone concentrations in PRD. As one of many possible effects of humidity, ozone may be more destroyed through
the photolysis reaction of $O_3 + hv \rightarrow O(^1D) + O_2$ as $O(^1D)$ can subsequently react with $H_2O$, forming OH through reaction of
$O(^1D) + H_2O \rightarrow 2OH$, which will be enhanced in environments with high humidity (Wang et al., 2013; Young et al., 2013).
In addition, despite more OH may be available given high humidity, OH can react with $NO_2$, forming $HNO_3$ in highly $NO_x$-
polluted regions, which ultimately leading to less efficient $O_3$ formation by competing with the oxidation of VOC and CO with
OH (Lu et al., 2019a). The negative correlation between humidity and ozone in PRD region has been identified by previous
studies (W. Zhang et al., 2021; Yang et al., 2021; Hua et al., 2008), and the high humidity environment in southern China may
be the result of moisture marine air masses transported from tropical region, South China Sea and western Pacific (W. Zhang
et al., 2021; Ding and Chen, 2005). For a non-linear learning framework using RFR, the second most important meteorological



driver in PRD is again the level of surface solar radiation. Interestingly, meridional wind at 850 hPa is key to ozone occurrence
in PRD, and it is negatively correlated with average MDA8 ozone. More generally, the regional average of MDA8 ozone in
PRD is negatively correlated with meridional wind at 850hPa from South China Sea (Fig. 6b), indicating strong marine air
inflow may have a significant cleaning and dispersion effect on PRD ozone and its precursors. Furthermore, the negative
correlation also expands to the northeast areas to the PRD, suggesting lower ozone in PRD given strong southerly wind in
these areas, which may hinder the transport of ozone and its precursors to PRD. This finding is consistent with the backward
trajectories and numerical modelling analysis by Qu et al. (2021).

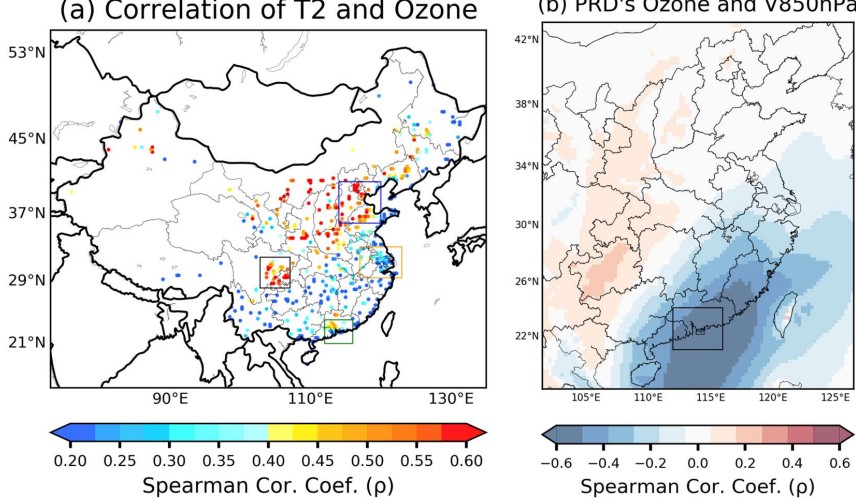


**Figure. 6 (a) Spearman correlation between daytime (06:00 to 18:00) averaged temperature at 2 m and MDA8 ozone from 2015 to 2019 from April to October. (b) Correlation coefficients between regional average of MDA8 ozone in PRD and daytime meridional wind at 850hPa at each ERA5 grid point from April to October of 2015 to 2019. A positive value of meridional wind indicates southerly wind.**

Additionally, previous studies (Jiang et al., 2015; Z. Chen et al., 2021; Qu et al., 2021; Wei et al., 2016) also indicate the
importance of vertical downward transport of ozone in southern region of China due to typhoons. The geographical location
and the intensity of typhoons can modulate the level of ozone in PRD; when typhoons are located relatively far away from
PRD during their development period, ozone can be elevated by downward movement of air masses, atmospheric stagnation
and lower planetary boundary layer height (Z. Chen et al., 2021), leading to suppressed dispersion of ozone and its precursors
before typhoon landing (Jiang et al., 2015; Z. Chen et al., 2021).
To illustrate the importance of such larger-scale meteorological effects on ozone pollution in PRD, we refer back to our
two-dimensional (2D) approach for RFR in PRD region first introduced and described in Sect. 3.2. We show the Gini feature
importance scores for this 2D domain approach in Fig. 7(a). Since we have multiple feature importance for each meteorological



variable in this set-up (i.e., one for each grid point in the 2D predictor domain), we sum up Gini scores for all grid points within
the expanded domain for each meteorological variable; and this summed value is denoted as the importance for that particular
meteorological variable. As illustrated in Fig. 7 (a), the relative feature importance of vertical velocity at 850hPa (w850)
increases compared to RFR using only local predictors (see Fig. 5b), likely reflecting the larger-scale influences of downward
transport of air masses in PRD region. Other key meteorological drivers (RH, solar radiation and meridional wind at 850hPa)
remain in a similar order to what was identified by purely local regressions. The model performance is slightly improved by
adding the 2D information with an increase of $R^2$ to 0.73 (from 0.70) in comparison to original RFR without 2D expansion.
However, we note that there appears to be a trade-off between the inclusion of non-linear relationships using RFR and
collinearity in high dimensions. Indeed, we find that the $R^2$ in RFR-2D for PRD region (see Fig. 7b) is still slightly less than
the $R^2$ using RR-2D (0.76) and the predictions from RR-2D are closer to observations with less deviations at both high and
low ozone value predictions (see Fig. 4j), suggesting that RR is better at handling the dimensionality increase of predictors,
which now slightly outweighs the importance of non-linearity in high dimensions.

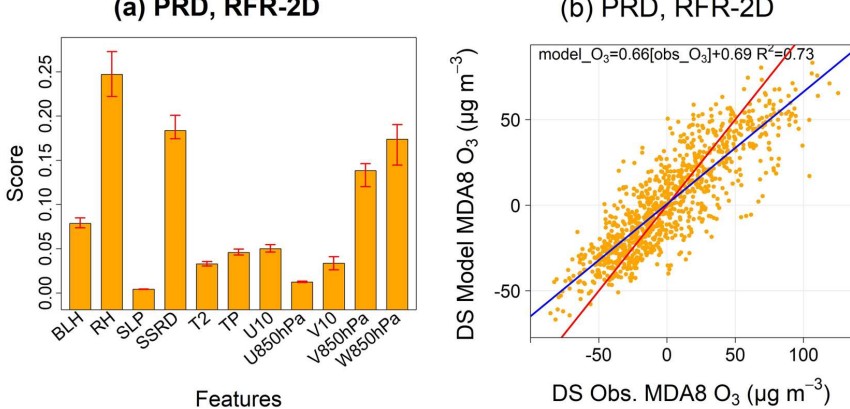


**Figure 7. (a) Average PRD Gini feature importance score of each meteorological variable if each regression includes non-local predictors within a 7.5° longitude × 7.5° latitude grid; the bar representations are consistent with Figure 5. (b) Linear fit between model prediction and observation in PRD using this 2D approach is drawn in blue line, red line equals the ideal 1:1 line.**

Across China, we found that there is a consistency in identification of the most important meteorological drivers by RFR
and RR. Temperature, solar radiation and RH are the three most commonly found most important meteorological drivers across
China, and the spatial distributions of these drivers are presented in Fig.8. Overall, there is a distinctive distribution pattern of
the 3 major meteorological drivers in China. Temperature at 2 m is dominant over northeast China, covering BTH and expand
to the norther region of China. Most areas in mid-latitude region of China including East China (e.g., YRD) and Sichuan Basin
show solar radiation as the main meteorological driver for ozone. The dominance of solar radiation gradually expands
northward and southward from this region while being overtaken by temperature in the north and relative humidity in the south.



Ozone in southern China is primarily driven by relative humidity. Such a distinctive spatial distribution of meteorological
drivers may be related to the characteristics of regional climatology. For instance, as previously described, regions in the
southern China such as PRD are more influenced by the moisture air masses, leading to the importance of humidity surpassing
temperature and solar radiation. While the relative drier northern regions do not have such a changeable humidity, making
temperature and solar radiation the key meteorological factors driving ozone.

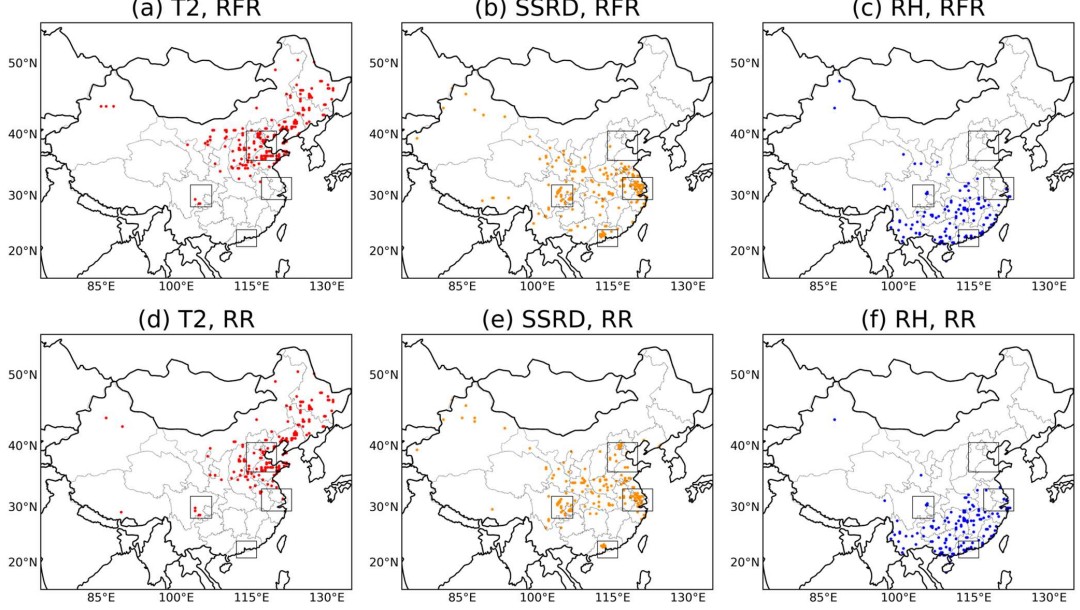


**Figure 8 (a)-(c) Most important meteorological drivers at each grid location from April to October of 2015 to 2019 as identified by**
**Gini importance using RFR. (d)-(f) The same but using absolute magnitudes for the slopes of RR. Variables as labelled. Relative**
**humidity (RH) dominates in the South and South-East, surface solar downward radiation (SSRD) primarily in the Central China**
**and Eastern China, and temperature at 2 m (T2) in the North and North-East China.**
**3.5 Anthropogenic and meteorological contributions to surface ozone trends from 2015 to 2019**
Finally, we explore how our new machine learning approach could be used to study the quantitative influence of
meteorology on historical ozone variability and trends in China. To facilitate a comparison to previous work, we use a similar
method as Li et al. (2020) to establish estimates for observed surface ozone trends in China. We note that our exercise is
somewhat limited by the slightly shorter period considered here, i.e., from 2015 to 2019, instead of starting from earlier years.
Given this very short period, we are aware that any long-term trend analysis is explorative and has to be interpreted carefully,
as will also become evident from low statistical significance in many detected trends. We nevertheless attempt such an analysis
to demonstrate how our method can be used in such contexts and to also evaluate if any statistically significant trends are



robust after accounting for meteorological influences. After all, as we have demonstrated above, we can quantify such
influences with greater skill than using simple MLR methods applied previously.

For trend analyses, we first convert MDA8 ozone concentrations from mass concentrations ($\mu g\ m^{-3}$) to volume mixing

ratios (ppbv). We then average MDA8 ozone over April to October or summertime for each year for both observational data
and model results predicted by our three best-performing controlling factor regressions (RFR, RR and RR-2D). The predictions
can be considered as a quantitative estimate for the influence of meteorology on the ozone record during the study period. The
residual (true ozone signal minus meteorological predictions) will for example be mainly reflective of anthropogenic
contribution but will also inevitably contain some uncertainties related to the accuracy of the machine learning algorithms in
modelling ozone.

Table 3 summarizes the regionally averaged observed trends from 2015 to 2019, which is estimated by ordinary linear

regression in the four regions. We additionally list our meteorologically estimated trends and the residual trends. Overall, the
three machine learning methods provide very similar estimates of meteorologically driven trends in BTH, YRD and Sichuan
Basin, while we find indications that the meteorologically driven trend in PRD may be underestimated by only using local
meteorological factors; using RR-2D we estimate a meteorologically driven trend of 0.84 ppbv $a^{-1}$ during April to October
from 2015 to 2019, while RFR and RR with only local meteorological predictors provide estimates of 0.1 ppbv $a^{-1}$ and 0.003
ppbv $a^{-1}$, respectively. Given the better prediction skill in RR-2D for this region (see Table 2 and Fig. 4), this further suggests
the importance of spatial meteorological phenomena for ozone trend attribution exercises in the PRD region.

In terms of the raw observed trends, both BTH and PRD show significant increases in ozone pollution ($p<0.05$) during

April to October from 2015 to 2019. We note that the observed trend in PRD is only significant if the months April to October
are considered, whereas there is no significant trend ($p=0.93$) if only examining months in summertime (JJA). This may be
attributed to the ozone's seasonality in PRD where highest ozone pollution occurs during autumn and the particularly high
ozone anomaly during September and October in 2019 (Fig. S6b). We underline that anthropogenic contribution (i.e., the
residual) may be overestimated in PRD if only local meteorological factors are considered, given that both residuals of RFR
and RR increase compared to RR-2D. For BTH, the positive ozone trend is found to be more significant during summertime
at 3.20 ppbv $a^{-1}$ ($p=0.05$) than if the whole April to October period (2.53 ppbv $a^{-1}$, $p<0.05$) is considered. Moreover, estimated
by RFR, the meteorologically driven trend in BTH is also higher at 0.74 ppbv $a^{-1}$ ($p<0.1$) during summertime than if the whole
April to October period is considered (0.45 ppbv $a^{-1}$; $p=0.14$). The April-to-October residual trends in BTH estimated by all
three algorithms are all greater than 2 ppbv $a^{-1}$ ($p<0.1$), indicating an elevated importance of anthropogenic drivers in BTH.
There are no significant observed trends in YRD and Sichuan. However, meteorological factors in both of these regions appear
to make a stronger influence according to RFR, RR and RR-2D. In terms of regional averages, all three of the machine learning
algorithms also agree on meteorologically driven negative trends in Sichuan while positive trends are found for YRD.
**Table 3. Observational, meteorological and residual trends of regional averaged MDA8 ozone (ppbv $a^{-1}$) from 2015 to 2019 for both**
**April to October and Northern Hemisphere summertime (June, July, August). Values within the brackets are the $p$ values for each**
**trend. Trends and $p$ values are in bold given $p$ values smaller than 0.1.**



| Method | Regions | 2015-2019 Apr. to Oct. | | | 2015-2019 Summer | | |
|---|---|---|---|---|---|---|---|
| | | Observed | Meteorological | Residual | Observed | Meteorological | Residual |
| RFR | BTH | **2.53 (0.02)** | 0.45 (0.14) | **2.08 (0.04)** | **3.2 (0.05)** | **0.74 (0.08)** | **2.46 (0.06)** |
| | PRD | **1.18 (0.02)** | 0.1 (0.88) | **1.08 (0.08)** | -0.12 (0.93) | -0.75 (0.14) | 0.64 (0.58) |
| | Sichuan | -0.34 (0.57) | **-0.75 (0.04)** | 0.42 (0.32) | 0.01 (0.99) | -0.91 (0.34) | 0.92 (0.11) |
| | YRD | 0.87 (0.36) | **1.38 (0.04)** | -0.51 (0.48) | 1.53 (0.15) | **1.35 (0.07)** | 0.17 (0.81) |
| RR | BTH | **2.53 (0.02)** | 0.37 (0.17) | **2.17 (0.03)** | **3.2 (0.05)** | 0.54 (0.18) | **2.66 (0.05)** |
| | PRD | **1.18 (0.02)** | 0.003 (0.997) | **1.18 (0.09)** | -0.12 (0.93) | -1.13 (0.11) | 1.01 (0.39) |
| | Sichuan | -0.34 (0.57) | **-0.84 (0.05)** | 0.51 (0.18) | 0.01 (0.99) | -0.84 (0.4) | **0.85 (0.06)** |
| | YRD | 0.87 (0.36) | **1.41 (0.04)** | -0.54 (0.43) | 1.53 (0.15) | **1.38 (0.09)** | 0.14 (0.86) |
| RR-2D | BTH | **2.53 (0.02)** | 0.47 (0.35) | **2.06 (0.09)** | **3.2 (0.05)** | 0.7 (0.33) | 2.5 (0.11) |
| | PRD | **1.18 (0.02)** | 0.84 (0.31) | 0.34 (0.58) | -0.12 (0.93) | -0.33 (0.62) | 0.21 (0.81) |
| | Sichuan | -0.34 (0.57) | **-0.86 (0.02)** | 0.52 (0.25) | 0.01 (0.99) | -0.68 (0.46) | 0.69 (0.21) |
| | YRD | 0.87 (0.36) | **1.45 (0.08)** | -0.58 (0.47) | 1.53 (0.15) | **1.63 (0.02)** | -0.10 (0.91) |


Finally, we aim to calculate trends on a ERA5 grid-by-grid point basis. Although RFR, RR and RR-2D all show significant
skill in modelling ozone across China, RR-2D exhibited particularly increased predictive skill in southern China. Therefore,
for assessing meteorologically-driven trends of MDA8 ozone across all ERA5 grid locations in China, we will only be
examining the results for RR-2D. Fig. 9 shows trends during April to October from 2015 to 2019 across China. Overall, the
observed average trend across China is 1.1 ppbv a$^{-1}$. The meteorologically driven trend of RR-2D gives the average at 0.5 ppbv
a$^{-1}$ across China, which is around 45% of the total trend. From Fig. 9 (a), most regions in eastern China show a positive trend
and the magnitudes of increase are more apparent in areas within and nearby BTH, where the ozone pollution increased at an
average rate of 2.6ppbv a$^{-1}$ across all grids within BTH. We find that the positive trend in those particular regions may be less
driven by meteorological factors but indeed might be the result of anthropogenic influences on air pollution (e.g., Liu and
Wang, 2020). In YRD, meteorologically driven positive trends are in general the highest in eastern China (average at 1.47
ppbv a$^{-1}$ across all grids in YRD), which is close to the regional averaged result by RR-2D (1.45 ppbv a$^{-1}$, *p*=0.08) in Table 3.
Observed trends in Sichuan are a mixture of both increases and decreases, but meteorologically driven trends are all negative
within this region. In PRD, meteorological factors likely played a more central role in driving the recent positive trends in
ozone pollution according to our analysis.

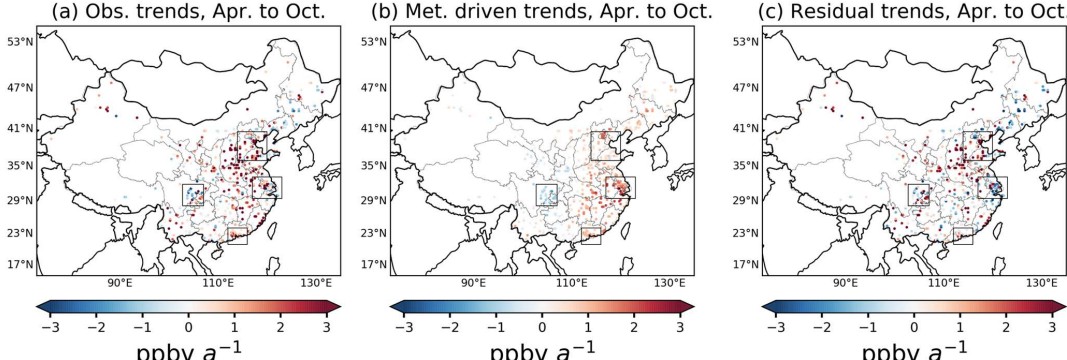


**Figure. 9 Trends of MDA8 ozone during April to October from 2015 to 2019. (a) shows the observed trend. (b) shows the mean
meteorologically driven trends of MDA8 ozone according to RR-2D. (c) shows the residual (approximating anthropogenic effects).
The trends are estimated by the slope of an ordinary linear regression fitting each year's April-October MDA8 average ozone values
from 2015 to 2019.**

## 4 Conclusion

Ozone pollution in China can be strongly influenced by meteorological conditions. This study examines the major
meteorological drivers for ozone pollution across China during months with particularly high ozone pollution (i.e., April to
October, from 2015 to 2019) using a controlling factor framework and two machine learning algorithms, namely random forest
regression (RFR) and ridge regression (RR).

The results obtained with RFR and RR are also compared with conventional approaches i.e., multiple linear regression
(MLR) and stepwise MLR, using consistent out-of-sample cross-validation methods. When considering local meteorological
factors only, RFR outperforms the linear approaches RR and MLR, which in turn perform better than stepwise MLR that uses
the only the three locally most significant meteorological factors. A major advantage of RFR is its ability to model non-linear
relationships (e.g., often observed between temperature and ozone). In addition, we tested how the consideration of larger scale
meteorological controlling factors improves our predictive performance. MLR noticeably suffers from the "curse of
dimensionality" due to the large increase of the number of predictors when we included additional meteorological information
spanning a 7.5°×7.5° domain around the target grid point for ozone pollution. In contrast, RR can deal well with this increase
in the number of predictors subject to an objective cross-validation approach for its hyperparameter tuning. In particular,
despite not directly considering non-linearity, we find an improvement of model performance in RR with additional 2-
dimensional predictors, which outperforms RFR with only local meteorological predictors in southern China, indicating the
importance of considering a wider meteorological context in future controlling factor analyses of this kind.

A key advantage of our approach is that both RFR and RR allow for a straightforward interpretation of the predictive
models. Reassuringly, we find a good agreement regarding the identification of the dominant local meteorological drivers for



each region. In general, ozone pollution in northern China such as in the Beijing-Tianjin-Hebei (BTH) region is predominantly
driven by temperature fluctuations while ozone in southern China like in Pearl River Delta (PRD) region is particularly strongly
controlled by humidity, possibly due to the important role of humid weather in preventing significant ozone pollution episodes
in this region, while the effect of humidity is constrained in BTH probably because of the relatively drier condition in this
region. Besides, we observe a strong influence in PRD of air exchange with pristine marine regions, leading to a greater
influence of large-scale wind directions, e.g., through the transport of clean marine air into the region, or through air stagnation
and ozone accumulation under large-scale sinking atmospheric motion. Surface solar radiation plays a major role in general
due to its importance for setting the conditions for ozone photochemistry, which is particularly dominant in the Yangtze River
Delta (YRD) and Sichuan Basin. Our work thus highlights that surface solar radiation might be a key predictor to consider in
future controlling factor analyses in these two regions. In summary, hot, dry and sunny weather tends to provide more favorable
conditions for ozone pollution in China, which is not entirely unexpected but carries important implications for future changes
in air pollution under climate change, while simultaneously considering the pivotal role of targeted emission control strategies
on ozone precursors.

In terms of ozone trends, we find a linear MDA8 ozone increase of about 1.1 ppbv $a^{-1}$ on average during April to October

from 2015 to 2019 across China. Regionally, these trends can be more than twice as large as in BTH. The largest positive
trends may be mostly attributed to non-meteorological factors such as change of precursors' emissions. However,
meteorologically driven trends on average shows increases at 0.5 ppbv $a^{-1}$ across China, equalling almost 50% over the period
considered here, and it is thus estimated to be more important factor, especially in southern China and the YRD region. While
the effect of meteorology might generally hinder extreme ozone pollution in the Sichuan Basin region, we conclude that
maintaining continuous emission control strategies is preferable in case of the occurrence of unfavorable weather conditions
for ozone mitigation.

**Data/Code availability**

The original air quality data including hourly and 8-hour rolling mean of ozone are available at https://quotsoft.net/air/

(Wang, X. L., 2021; last accessed: 13 July 2021). The ERA5 reanalysis product is available at
https://cds.climate.copernicus.eu/ (last accessed: 11 November 2021). The codes for machine learning algorithms are available
from the corresponding author upon request.

**Author contribution**

P.N., X.W. and G.F. designed the study. X.W. performed the modelling and analysis of the data, supervised by P.N. and

G.F. X.W. wrote the paper with input and revision from P.N. and G.F.



**Competing interests**
The authors declare that they have no conflict of interest.

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
