# Peer review of "A machine learning approach to quantify meteorological drivers of 2015-2019 ozone pollution in China"

_Atmospheric Chemistry and Physics, 2021_

## Author Response (AR1)

**Responses to Reviewer #2**

We thank the reviewer very much for the supportive and constructive comments, which have helped us to further improve our manuscript.

Below we reply point-by-point to the reviewer's comments (black) in blue font. The specific lines we are referring to here are from the manuscript with tracked changes.

The authors applied advanced statistical approaches to identify major meteorological drivers of ozone pollution over China. They also compared their results with the multiple linear regression methods. Moreover, they also found that by including the large-scale meteorology, their model skill will be improved relative to the model constructed by only local meteorological variables. Based on these regression models, the authors demonstrate the models' capability and advantage in the understanding of major meteorological drivers of ozone pollution and in the isolation of meteorological effects from observed ozone trends.

Ozone pollution issue in China is of great concern. This study adds insights into the better understanding of recent ozone trend in China. I think the major novelty of this work is its new methods. However, several places should be improved in order to highlight this strength. Please find my comments below.

The Abstract should be revised. Firstly, I am surprised that there are almost no quantitative sentences to show the advantages of the machine learning approach. It prevents the readers from easily understanding the contribution of this work in current version.

Thank you for the suggestion. We agree that highlighting a selection of quantitative improvements using machine learning algorithms is helpful to underline the significance of our work. Therefore, we have added information to lines 20 to 21 in the abstract specifying the improved coefficient of determination ($R^2$) using RFR:
"as evident from its higher coefficients of determination ($R^2$) with observations (0.5 to 0.6 across China) when compared to the linear methods (typically $R^2$=0.4-0.5)".

Specifically, given the context of only using local meteorological predictors, we find that most $R^2$ values using non-linear RFR range from 0.5 to 0.6 across China, while the three linear regressions (i.e., RR, MLR and stepwise MLR) show lower $R^2$ score ranges (mostly from 0.4 to 0.5). We now also mention the quantitative improvement for RR when using patterns of non-local meteorological predictors, specifically for southern China where averaged $R^2$ increases from 0.47 to 0.6 (lines 23 to 24).

Then, a large fraction of the Abstract is the description of leading meteorological

variables. However, these results are not new and have been reported by a lot of studies previously.

Section 3.4. As I mentioned above, this section shows a lot of previously-reported knowledge on the major meteorological drivers of ozone pollution. I suggest the authors to make this concise and to highlight your new findings.

We agree that improving conciseness is key, especially given the additions to the abstract in response to the comment above. We have thus removed several sentences (lines 27 to 31) from the abstract and now more clearly highlight how our results differ from previous studies. For instance, we find that inclusion of non-local meteorological predictors can not only improve the prediction skill of RR but also more accurately reflect the higher meteorological contribution to ozone increase trend in southern China (line 24 to 25):
"Moreover, this improved RR shows a higher averaged meteorological contribution to the increase trend of ozone pollution in that region, pointing towards an elevated importance of large-scale meteorological phenomena for ozone pollution in southern China."

Besides, we find that including variable of surface solar radiation is important for controlling factor analyses as this is one of the key meteorological factors for ozone pollution across China. We still also note – more briefly – how our study is validated through results matching previous works and expected meteorology-ozone relationships (i.e., lines 31 to 33):
"In line with expectations, our analysis underlines that hot and dry weather conditions with high sunlight intensity are strongly related to high ozone pollution across China, thus further validating our novel approach."

We have further addressed the related comment concerning section 3.4. Specifically, we have condensed and summarized lines 406 to 414 into a single sentence (lines 402 to 406). This text describes the importance of temperature for ozone pollution in the BTH region and suggests possible mechanisms for which increased temperature could affect ozone production. We have decided to discard the detailed explanation of how typhoons may affect PRD's ozone (lines 453 to 457) and we have replaced this with lines 451 to 453 to point out the limitations of using regressions with local predictors as an emphasis for the importance of capturing larger-scale meteorological phenomena.

Section 3.5. Another contribution of this work is the quantification of meteorological role in ozone trends. However, I failed to find the comparison between machine learning method and MLR method in this Section. At least, I suggest the authors to list the MLR-based estimates in Table 3.

Thank you for this helpful feedback. We have added the meteorologically driven trends estimated by MLR using 11 local meteorological predictors to Table 3. We find that

these trends are similar to the trends estimated by RR. This is mostly due to the similar model performances of these two linear regression methods (Table 2). However, we leave out stepwise MLR and MLR-2D in this section because of the overall weak performance of these methods in predicting ozone (Table 2), which could lead to false/misleading interpretations of our results.

The "recent" in the Title is not clear. It is better to be replaced by "2015-2019".

We agree and have changed the title to "A machine learning approach to quantify meteorological drivers of 2015-2019 ozone pollution in China".

L41: Brief information on VOC emission changes should be added.

Thanks. We have reorganized the first two paragraphs and have added some information to reflect the changes of VOC emissions from (lines 62 to 63) based on the estimates reported by Zheng et al. (2018):
"Notably, the total emissions of nonmethane volatile organic compounds (NMVOCs) have actually increased by 11% in 2017 compared to 2010 (Zheng et al., 2018)."

In addition, we have made some changes and adjustments as follows:

(1) In lines 467 to 477, we have re-examined the difference of prediction skill between RFR-2D and RR-2D in PRD and found that the higher $R^2$ from RR-2D may be mostly attributed to its ability of better extrapolating extreme high/low anomalies of observed ozone, while RFR-2D may be less capable in this respect since its prediction range is more subjected to the range of training data. However, the seemingly better slope (i.e., closer to 1) of RR-2D may be due to the tilt of linear fit with a higher slope, which is caused by its limitation of over-extrapolation. We provided the time series of predicted ozone by RFR-2D and RR-2D to Fig. S6 in the supplementary material as examples for further illustrating our interpretation. In addition, we have adjusted the scale of y-axis in Fig. 7b to the same scale of Fig. 4j for comparison.

[Figure]

Figure S6. Examples of deseasonalized ozone predicted by RR-2D (a) and RFR-2D (b) in comparison with deseasonalized observations in PRD during April to October of 2015. For the low anomaly on 2015-Oct-4 (indicated by the black arrow in the figure), RR-2D has a better prediction compared to RFR-2D, which suggests its ability of extrapolation; while the overprediction of high anomaly by RR-2D on 2015-Apr-14 (red arrow) indicates its trade-off for having a risk of over-extrapolation.

[Figure]

Figure 7. (a) Average PRD Gini feature importance score of each meteorological variable if the RFR regressions include non-local predictors within a 7.5° longitude × 7.5° latitude domain around the predicted grid point; the bar representations are consistent with Figure 5. (b) Linear fit between RFR-2D predictions and observations in PRD (blue line). The red line equals the ideal 1:1 relationship.

(2) In lines 261 to 264, we have clarified the definition of $R^2$ which is used as a metric for assessing the prediction skills of all algorithms:

"Therefore, we adopt the coefficient of determination ($R^2$) as a standard metric for the evaluation of prediction performance, which assesses the goodness-of-fit for the linear regression between the deseasonalized MDA8 ozone data and the predicted values (e.g., Han et al., 2020)."

We have enlarged the font sizes for both the equations of linear regressions and $R^2$ in Fig. 4.

[Figure]

**Figure 4 (a)-(d) Comparison of regional averages of deseasonalized MDA8 ozone between model predictions and observations for RFR, (e)-(h) RR and (i)-(l) RR-2D. Linear fits between predicted and observed data are indicated by blue lines; red lines are the ideal 1:1 lines. The values for both models and observations are averaged over all ERA grid points in each region. Each graph contains information of the linear regression with slope and $R^2$ value (coefficient of determination).**

(3) In lines 65 to 66, we have added text to highlight that there is still uncertainty regarding the increases in ozone by the effect of HO₂ uptake on aerosol, and we have cited the study of Tan et al. (2020) as an example for describing such an uncertainty:

"the quantitative contribution to the increases of ozone from $HO_2$ uptake on aerosol remains uncertain (e.g., Tan et al., 2020)"

(4) We have added further information with regards to the ranges of $R^2$ from the three linear regressions with local predictors (i.e., RR, MLR and stepwise MLR) in lines 284 to 287:
"Although most $R^2$ values across China for these three linear regressions (i.e., RR, MLR and stepwise MLR) are within the same range of 0.4 to 0.5, stepwise MLR shows the worst performance with consistently lower $R^2$ values across China, and more of these values fall in a lower range of 0.3 to 0.4."

(5) In the conclusions (lines 583 to 587), similar to the abstract, we have provided the ranges of $R^2$ values to quantitively compare the modelling skill of RFR and the other three linear regressions (i.e., RR, MLR and stepwise MLR):
"The better performance of RFR is for example evident from the overall increase in predicted versus observed coefficients of determination ($R^2$) ranging from 0.5 to 0.6, as compared to 0.4 to 0.5 for the three linear regressions. Stepwise MLR attains the lowest averaged $R^2$ of all these methods across China."

(6) We have replaced the abbreviation for vertical velocity at 850 hPa from W850 to W850hPa throughout whole manuscript including Fig. 5 as to make it consistent with the naming of variables U850hPa and V850hPa.

[Figure]

Figure 5 (a)-(d) Average Gini feature importance scores of each meteorological variable for RFR in each region. (e)-(h) Average slopes of each meteorological variable for RR in each region. The red bars indicate the range of importance scores/slopes found across the five regression models learned to predict the left-out test years.

(7) We have made an adjustment to Fig. 6 (b): we now show the correlation coefficients between regional average of MDA8 ozone in PRD and meridional wind at 850 hPa. Previously, both deseasonalized MDA8 ozone in PRD and

deseasonalized meridional wind at 850 hPa were used in this figure. This was not consistent with the description of the figure in the main text. Note that this visualization issue does not affect our conclusions or overall results.

[Figure]

Figure. 6 (a) Spearman correlation between daytime (06:00 to 18:00) averaged temperature at 2 m and MDA8 ozone from 2015 to 2019 from April to October. (b) Correlation coefficients between the regional average of MDA8 ozone in PRD and the daytime (06:00 to 12:00) meridional wind at 850 hPa at each ERA5 grid point from April to October of 2015 to 2019. A positive value of meridional wind indicates southerly wind.

**References**

Han, H., Liu, J., Shu, L., Wang, T., and Yuan, H.: Local and synoptic meteorological influences on daily variability in summertime surface ozone in eastern China, Atmos. Chem. Phys., 20, 203–222, https://doi.org/10.5194/acp-20-203-2020, 2020.

Tan, Z., Hofzumahaus, A., Lu, K., Brown, S. S., Holland, F., Huey, L. G., Kiendler-Scharr, A., Li, X., Liu, X., Ma, N., Min, K. E., Rohrer, F., Shao, M., Wahner, A., Wang, Y., Wiedensohler, A., Wu, Y., Wu, Z., Zeng, L., Zhang, Y., and Fuchs, H.: No Evidence for a Significant Impact of Heterogeneous Chemistry on Radical Concentrations in the North China Plain in Summer 2014, Environ. Sci. Technol., 54, 5973–5979, https://doi.org/10.1021/acs.est.0c00525, 2020.

Zheng, B., Tong, D., Li, M., Liu, F., Hong, C., Geng, G., Li, H., Li, X., Peng, L., Qi, J., Yan, L., Zhang, Y., Zhao, H., Zheng, Y., He, K., and Zhang, Q.: Trends in China's anthropogenic emissions since 2010 as the consequence of clean air actions, Atmos. Chem. Phys., 18, 14095–14111, https://doi.org/10.5194/acp-18-14095-2018, 2018.